# Fine-Tuning Generative Models as an Inference Method for Robotic Tasks

**Orr Krupnik**          **Elisei Shafer**          **Tom Jurgenson**          **Aviv Tamar**

Department of Electrical and Computer Engineering
Technion - Israel Institute of Technology
Haifa, Israel
`ok1@campus.technion.ac.il`

**Abstract:** Adaptable models could greatly benefit robotic agents operating in the real world, allowing them to deal with novel and varying conditions. While approaches such as Bayesian inference are well-studied frameworks for adapting models to evidence, we build on recent advances in deep generative models which have greatly affected many areas of robotics. Harnessing modern GPU acceleration, we investigate how to quickly adapt the sample generation of neural network models to observations in robotic tasks. We propose a simple and general method that is applicable to various deep generative models and robotic environments. The key idea is to quickly fine-tune the model by fitting it to generated samples matching the observed evidence, using the cross-entropy method. We show that our method can be applied to both autoregressive models and variational autoencoders, and demonstrate its usability in object shape inference from grasping, inverse kinematics calculation, and point cloud completion.

## 1  Introduction

Humans and other animals maintain powerful mental models of the world [1] for navigation, object manipulation, social interaction and other day-to-day tasks. These mental models are imperfect and tend to be inaccurate. However, they are highly adaptable and are frequently updated upon arrival of new information. Robotic agents operating in diverse and unstructured environments may also be required to adjust their behavior, and would therefore benefit from adaptable models similar to the ones humans hold. Recent studies in robot manipulation and navigation have focused on scenarios where an accurate model can be learned and used as-is in downstream planning tasks [2, 3, 4]. In contrast, inspired by biological mental models, this paper focuses on how to efficiently *adapt* a model to novel information.

Natural approaches to updating models given new evidence such as *Bayesian inference* have been used extensively in computer vision and robotics [5, 6]. When the modeled probability distributions are low-dimensional, Bayesian inference may have closed form solutions or the posterior can be numerically estimated using methods such as Markov chain Monte Carlo (MCMC, [7]). However, there is growing interest in using high-dimensional *deep generative models* for robotics, which can represent complex and diverse data. Advances such as *variational autoencoders* [8] and *diffusion models* [9] make it possible to learn diverse and high-dimensional distributions. For such expressive models, Bayesian inference techniques are not suitable, and they cannot operate on the time-scale required for robotic tasks [10]. Another approach, which has been found effective with deep generative models, is to *train* them to adapt by conditioning on possible observations [11, 12]. However, this paradigm may fall short when faced with out-of-distribution evidence at test-time.

We propose a simple approach for updating the parameters of deep generative models given empirical observations, to approximate complex posterior distributions. Our method requires a forward

7th Conference on Robot Learning (CoRL 2023), Atlanta, USA.

simulation of the environment which can produce observations given a model, and a similarity function for observations. We build on GPU-based physics simulation [13] and model training to perform fast inference in a novel robotic scenario by fine-tuning the generative model weights. Considering the parametric nature of deep generative models, we use a version of the cross-entropy method (CEM, [14]) to quickly update the parameters to generate observations conforming with the available evidence. We dub our method MACE, for **M**odel **A**daptation with the **C**ross-**E**ntropy method.

To showcase MACE, we focus on robotic domains where the forward simulation step is easily performed using off-the-shelf physical simulators. We demonstrate the versatility of MACE on several robotic manipulation tasks that we frame as model adaptation, using two different types of deep generative models. In particular, we demonstrate results on object identification from position measurements of a multi-fingered robot gripper, on recovery of object point clouds given partial measurements (as generated by depth sensors) and on an inverse kinematics (IK) task in the presence of obstacles. In all of these environments, the posterior has a rich multi-modal structure. We demonstrate that MACE is indeed capable of producing diverse posterior samples for a variety of observations, and that it outperforms baseline approaches in diversity and accuracy. In terms of speed, we show that by exploiting GPU-based simulation and inference, our fine-tuning can be performed online in a competitive time frame. For example, in the IK tasks, we find that MACE can outperform the MoveIt [15] library in quickly finding IK solutions for complex scenarios.

## 2 Related Work

*Bayesian inference* is concerned with computing the posterior distribution of data given observations. Exact computation is possible for simple models which admit conjugate priors, and Markov chain Monte Carlo (MCMC) can be used for general models [7]. Approximate Bayesian computation (ABC, [16]) allows sampling from the posterior without an exact likelihood, but with some similarity function between observations. MACE is inspired by these approaches, and expedites the search for better samples using CEM [14]. Recent work by Engel et al. [17] introduces a Bayesian model update scheme using CEM related to ours. However, their method requires the true likelihood and is limited to relatively small models, while ours can be applied to deep generative models.

Bayesian inference has been used extensively in robotics in the context of state estimation, localization, and mapping [5, 18]. In recent work, Marlier et al. [19] use Bayesian estimation of a posterior distribution of grasp poses for multi-finger object grasping, and Pastor et al. [10] use Bayesian inference with LSTM [20] to classify objects using tactile sensors. Both the above are application-specific, while MACE is a general approach applicable to a variety of tasks and generative models.

Another approach is to amortize inference by learning an approximate posterior using data from the joint distribution of models and observations $p(\boldsymbol{x}, \boldsymbol{o})$ such as in *conditional variational autoencoders* (CVAE, [21]). We compare MACE with a CVAE baseline, and show that while our inference procedure is slower, MACE produces a more diverse posterior. Furthermore, MACE can work for observations that are out of distribution with respect to $p(\boldsymbol{x}, \boldsymbol{o})$, unlike the CVAE. Finally, MACE can tune the same prior model with different modalities of observations without retraining.

*Meta-learning*, and *meta-RL* in particular, is an alternative approach to quickly adapt behavior to new evidence [22]. However, meta-RL is typically model-free, and learns how to adapt a policy [22, 23]. Model-based meta-RL approaches such as Zintgraf et al. [12] use a CVAE to condition on the history of observations. Whether such methods could be improved using our inference method is an interesting direction for future research.

## 3 Model Adaptation with the Cross-Entropy Method

We now describe MACE, our method for adapting deep generative models to environment observations using the cross-entropy method. We begin by describing the setup and the types of tasks we aim to solve (Sec. 3.1); next, we discuss our update rule and present the full algorithm (Sec. 3.2).

## 3.1 Problem Formulation

In the robotics context, an *inference problem* involves the recovery of *task* parameters given observations of the environment. We assume some distribution $p(\boldsymbol{x})$ over task representations $\boldsymbol{x} \in \mathcal{X}$. A task description $\boldsymbol{x}$ could be a point cloud (PC) of an object as in a grasping task; an image input for more complex manipulation; or a desired joint configuration in a reaching task. MACE requires access to a generative model representing a parametric distribution over task representations $p(\boldsymbol{x}; \boldsymbol{\theta})$. We assume the generative model is initially trained to represent the prior, i.e., for some initial parameter $\boldsymbol{\theta}_0$ it holds that $p(\boldsymbol{x}; \boldsymbol{\theta}_0) = p(\boldsymbol{x})$.

Although the task representation $\boldsymbol{x}$ may be unknown, some information about it can be observed. We denote this information $\boldsymbol{o} \in \mathcal{O}$ (for *observation*) and note that it can be of any form. For example, in a grasping task, $\boldsymbol{o}$ could be a partial PC obtained from a depth sensor (while $\boldsymbol{x}$ is the full object model); in a reaching task, $\boldsymbol{o}$ could represent obstacles for the robot to avoid.

A central component of MACE is a *simulator* of the task. Exact environment simulators are often hard to design, leaving gaps to the reality they attempt to simulate [18]. However, for our purposes, we only require the simulator to emit observations of a similar modality to the ones emitted by the environment. Therefore, the simulator can be viewed as a probability distribution $\hat{p}(\boldsymbol{o}|\boldsymbol{x})$, providing observations given task representations.

The goal of the inference task is to train the parametric model $p(\boldsymbol{x}; \boldsymbol{\theta})$ to produce a distribution closely resembling the posterior over task representations given observations, $p(\boldsymbol{x}|\boldsymbol{o})$.

## 3.2 Updating the Model

We aim to update the model parameters $\boldsymbol{\theta}$ so that the generative model $p(\boldsymbol{x}; \boldsymbol{\theta})$ more closely resembles the posterior $p(\boldsymbol{x}|\boldsymbol{o})$. We can do this by minimizing the Kullback-Leibler (KL) divergence between the posterior and the parametric model:

$$\arg\min_{\boldsymbol{\theta}} D_{\mathrm{KL}}\left(p(\boldsymbol{x}|\boldsymbol{o})\|p(\boldsymbol{x}; \boldsymbol{\theta})\right) = \arg\min_{\boldsymbol{\theta}} \int p(\boldsymbol{x}|\boldsymbol{o}) \log p(\boldsymbol{x}|\boldsymbol{o}) d\boldsymbol{x} - \int p(\boldsymbol{x}|\boldsymbol{o}) \log p(\boldsymbol{x}; \boldsymbol{\theta}) d\boldsymbol{x}.$$

The parametric model is only present in the second term, therefore we can maximize it to minimize the entire KL divergence. Since the posterior $p(\boldsymbol{x}|\boldsymbol{o})$ is unknown, we use Bayes' rule to replace it with the likelihood $p(\boldsymbol{o}|\boldsymbol{x})$: $\arg\max_{\boldsymbol{\theta}} \int p(\boldsymbol{x}|\boldsymbol{o}) \log p(\boldsymbol{x}; \boldsymbol{\theta}) d\boldsymbol{x} = \arg\max_{\boldsymbol{\theta}} \int \frac{p(\boldsymbol{o}|\boldsymbol{x})p(\boldsymbol{x})}{p(\boldsymbol{o})} \log p(\boldsymbol{x}; \boldsymbol{\theta}) d\boldsymbol{x} = \arg\max_{\boldsymbol{\theta}} \mathbb{E}_{\boldsymbol{x} \sim p(\boldsymbol{x})} \left[p(\boldsymbol{o}|\boldsymbol{x}) \log p(\boldsymbol{x}; \boldsymbol{\theta})\right].$

The likelihood of the observation $p(\boldsymbol{o}|\boldsymbol{x})$ is also an unknown quantity. However, we may approximate it using the forward simulator, which can produce observations $\boldsymbol{o}$ given $\boldsymbol{x}$. We define a *score function* as any function $S : \mathcal{O} \times \mathcal{O} \to [0, 1]$ indicating similarity between pairs of observations, and assume $\mathbb{E}_{\boldsymbol{o}' \sim \hat{p}(\boldsymbol{o}|\boldsymbol{x})} S(\boldsymbol{o}', \boldsymbol{o})$ is an approximation of $p(\boldsymbol{o}|\boldsymbol{x})$. An intuitive case to justify this assumption is when observations $\boldsymbol{o}$ are discrete, and $S(\boldsymbol{o}', \boldsymbol{o}) = \mathbf{1}_{\boldsymbol{o}'=\boldsymbol{o}}$ is an indicator of whether $\boldsymbol{o}'$ is equal to the evidence $\boldsymbol{o}$ [1]. When using a deterministic simulator, the expectation over observations $\mathbb{E}_{\boldsymbol{o}' \sim \hat{p}(\boldsymbol{o}|\boldsymbol{x})} S(\boldsymbol{o}', \boldsymbol{o})$ can be replaced by the score of a single sample $S(\boldsymbol{o}', \boldsymbol{o})$ [2]. Plugging in this score function, the optimization problem becomes:

$$\arg\max_{\boldsymbol{\theta}} \mathbb{E}_{\boldsymbol{x} \sim p(\boldsymbol{x})} \left[S(\boldsymbol{o}', \boldsymbol{o}) \log p(\boldsymbol{x}; \boldsymbol{\theta})\right]. \tag{1}$$

Recalling our assumption that $p(\boldsymbol{x}; \boldsymbol{\theta}_0) = p(\boldsymbol{x})$, Eq. 1 can be optimized using importance sampling: $\arg\max_{\boldsymbol{\theta}} \frac{1}{N} \sum_{i=1}^{N} \frac{p(\boldsymbol{x}_i; \boldsymbol{\theta}_0)}{p(\boldsymbol{x}_i; \boldsymbol{\theta})} S(\boldsymbol{o}_i, \boldsymbol{o}) \log p(\boldsymbol{x}_i; \boldsymbol{\theta})$, where $\boldsymbol{x}_i \sim p(\boldsymbol{x}; \boldsymbol{\theta})$, the sampling distribution, and $\boldsymbol{o}_i$ is the observation matching $\boldsymbol{x}_i$. One question is how to choose an effective sampling distribution which places enough mass on high-scoring $\boldsymbol{x}$ values. Inspired by the iterative approach of Engel et al. [17], we optimize this objective iteratively using stochastic gradient descent (SGD).

---

[1]For other examples of score functions, see the environment descriptions in Sec. 4.

[2]As the simulators in our experiments are deterministic, we simplify notation by using the single-sample score function. The derivation for stochastic simulators follows the same lines, using expected scores.

At each iteration, we use the previous parametric model as the sampling distribution and take a few gradient steps to obtain the next model parameters. The objective at iteration $t$ is given by:

$$\arg\max_{\boldsymbol{\theta}} \frac{1}{N} \sum_{i=1}^{N} \frac{p(\boldsymbol{x}_i; \boldsymbol{\theta}_0)}{p(\boldsymbol{x}_i; \boldsymbol{\theta}_{t-1})} S(\boldsymbol{o}_i, \boldsymbol{o}) \log p(\boldsymbol{x}_i; \boldsymbol{\theta}), \quad (2)$$

with $x_i \sim p(\boldsymbol{x}; \boldsymbol{\theta}_{t-1})$. In practice, we find that the importance sampling term in this objective makes it difficult to optimize due to large discrepancies between the parametric distributions[3]. Instead, we introduce another approximation and remove the importance sampling term $\frac{p(\boldsymbol{x})}{p(\boldsymbol{x};\boldsymbol{\theta})}$ from Eq. 2.

To further improve performance and shorten training times, we follow the cross-entropy method (CEM) formulation described by Botev et al. [24]. We view the objective in Eq. 1 as the problem of finding a distribution $p(\boldsymbol{x}; \boldsymbol{\theta})$ which produces samples with high scores $S(\boldsymbol{o}', \boldsymbol{o})$. To optimize $S(\boldsymbol{o}', \boldsymbol{o})$ (which is implicitly a function of $\boldsymbol{x}$ through the simulator), we treat $p(\boldsymbol{x}; \boldsymbol{\theta})$ as an importance sampling distribution and adjust it such that it samples values of $\boldsymbol{x}$ that are close to the ones implicitly maximizing $S(\boldsymbol{o}', \boldsymbol{o})$. At each iteration $t$ we sample a batch $\{\boldsymbol{x}_i \sim$

---

**Algorithm 1:** MACE (**M**odel **A**daptation with the **C**ross-**E**ntropy method)

**Input:** Prior model $p(\boldsymbol{x}; \boldsymbol{\theta}_0)$, simulator $\hat{p}(\boldsymbol{o}|\boldsymbol{x})$, observation $\boldsymbol{o}$, quantile parameter $q$, gradient steps parameter $M$

1 **for** $t \leftarrow 1, \ldots, T$ **do**
2     Sample $\boldsymbol{x}_1, \ldots, \boldsymbol{x}_N \sim p(\boldsymbol{x}; \boldsymbol{\theta}_{t-1})$
3     Obtain $\boldsymbol{o}_1, \ldots, \boldsymbol{o}_N \sim \hat{p}(\boldsymbol{o}|\boldsymbol{x})$ using the simulator
4     Calculate $S(\boldsymbol{o}_i, \boldsymbol{o})$ for all $\boldsymbol{o}_1, \ldots, \boldsymbol{o}_N$
5     Select top $qN$ samples, set $\delta = S(\boldsymbol{o}_{\lfloor qN \rfloor}, \boldsymbol{o})$
6     Optimize
    $\arg\max_{\boldsymbol{\theta}} \sum_{i=1}^{N} \mathbf{1}_{S(\boldsymbol{o}_i, \boldsymbol{o}) \geq \delta} \log p(\boldsymbol{x}; \boldsymbol{\theta}_t)$ using $M$ steps of SGD

7 **return** updated model $p(\boldsymbol{x}; \boldsymbol{\theta}_T)$

---

$p(\boldsymbol{x}; \boldsymbol{\theta}_{t-1})\}_{i=1}^{N}$, obtain observations using the simulator $\{\boldsymbol{o}_i \sim \hat{p}(\boldsymbol{o}|\boldsymbol{x}_i)\}_{i=1}^{N}$ and calculate their respective scores $\{S(\boldsymbol{o}_i, \boldsymbol{o})\}_{i=1}^{N}$. The top $qN$ samples with the best scores are selected, where $q$ is a pre-selected quantile hyperparameter. We define $\delta = S(\boldsymbol{o}_{\lfloor qN \rfloor}, \boldsymbol{o})$, the score function value for the $\lfloor qN \rfloor$-th sample. The complete CEM-inspired MACE objective is given by:

$$\arg\max_{\boldsymbol{\theta}} \sum_{i=1}^{N} \mathbf{1}_{S(\boldsymbol{o}_i, \boldsymbol{o}) \geq \delta} \log p(\boldsymbol{x}_i; \boldsymbol{\theta}). \quad (3)$$

Botev et al. [24] solve a stochastic program for each iteration of $\boldsymbol{\theta}$. Instead, we optimize this objective via stochastic SGD as described above. The full MACE algorithm is shown in Alg. 1.

### 3.2.1 Implementation

Although MACE is suitable in principle for any generative model, some considerations must be made for specific types of models.

**Autoregressive models** provide an explicit likelihood value for a sample $\boldsymbol{x}$ using the chain rule: $p(\boldsymbol{x}) = \prod_i p(x_i|x_{i-1}, \ldots, x_0)$. Therefore, they are straightforward to use with MACE by denoting the neural network weights as $\boldsymbol{\theta}$ and directly optimizing the objective in Eq. 3.

**VAEs** do not provide an explicit likelihood value $p(\boldsymbol{x})$. However, they are trained with a lower bound estimate of $p(\boldsymbol{x})$, namely the Evidence Lower Bound (ELBO) $\log p(\boldsymbol{x}|\boldsymbol{z}; \boldsymbol{\psi}) - D_{\mathrm{KL}}(q(\boldsymbol{z}|\boldsymbol{x}; \boldsymbol{\phi})\|p(\boldsymbol{z}))$, where $p(\boldsymbol{x}|\boldsymbol{z}; \boldsymbol{\psi})$ is the VAE decoder parameterized by $\boldsymbol{\psi}$, $q(\boldsymbol{z}|\boldsymbol{x}; \boldsymbol{\phi})$ is the encoder parameterized by $\boldsymbol{\phi}$ and $p(\boldsymbol{z})$ is the prior Gaussian distribution of the latent variables parameterized by $\boldsymbol{\mu_z}, \boldsymbol{\sigma_z}$. We can use the ELBO in Eq. 3 as a lower bound of the log-likelihood term $\log p(\boldsymbol{x}; \boldsymbol{\theta})$, optimizing the VAE parameters: $\arg\max_{\boldsymbol{\psi}, \boldsymbol{\phi}} \sum_{i=1}^{N} \mathbf{1}_{S(\boldsymbol{o}_i, \boldsymbol{o}) \geq \delta} [\log p(\boldsymbol{x}_i|\boldsymbol{z}_i; \boldsymbol{\psi}) - D_{\mathrm{KL}}(q(\boldsymbol{z}_i|\boldsymbol{x}_i; \boldsymbol{\phi})\|p(\boldsymbol{z}))]$. We found the optimization of the entire set of VAE weights $\boldsymbol{\psi}, \boldsymbol{\phi}$ to be difficult. Instead, we use $\boldsymbol{\theta} = \{\boldsymbol{\mu_z}, \boldsymbol{\sigma_z}\}$ as our tuned parameters, and keep the encoder and decoder weights frozen. This choice of $\boldsymbol{\theta}$ only affects the

---

[3]We compare MACE to this objective as a baseline; see experiment results in Sec. 4.1 and Sec. 4.2

second term of the objective, and specifically $p(\boldsymbol{z})$. The objective of this version of MACE, which we dub MACE-VAE, then becomes: $\arg\max_{\boldsymbol{\theta}} \sum_{i=1}^{N} \mathbf{1}_{S(\boldsymbol{o}_i,\boldsymbol{o})\geq\delta} \left[ -D_{\mathrm{KL}} \left( q(\boldsymbol{z}_i|\boldsymbol{x}_i)\|p(\boldsymbol{z};\boldsymbol{\theta}) \right) \right].$

## 4 Experiments

To demonstrate the potential of MACE in robotic applications, we consider several domains in which generative models are used to capture complex data distributions and show that MACE leads to practical solutions, improving on the alternatives. We describe two of these environments and how they fit our setting, followed by a summary of results. A third environment, in which we use MACE to recover object shapes from partial point cloud observations, is described in Appendix C.

### 4.1 Inferring Object Shapes by Grasping

Inferring object shapes from tactile measurements [10] is vital when visual sensors are unavailable or limited. We investigate whether details about an object can be inferred using only the contact points between it and the robot gripper fingertips. Differing from the approach of Pastor et al. [10], which aims to classify objects into 36 classes, we consider an expressive prior distribution over possible shapes represented by a deep generative model.

**Dataset.** We use the "airplane" class from the ShapeNet [25] dataset as a representative collection of objects with complex shapes. This class contains 4045 objects, represented as 2048-point PCs.

**Model.** The prior generative model $p(\boldsymbol{x};\boldsymbol{\theta}_0)$ is a VAE trained on full PCs of objects. We use MACE-VAE as described in Sec. 3.2.1. Our VAE architecture uses PointNet [26] in the encoder and a fully connected decoder, and is based on the implementation used by Daniel and Tamar [27]. Additional training details and hyperparameters can be found in Appendix A.

**Simulator.** For ease of implementation, we use a simple geometric simulator to calculate contact points. Details can be found in Appendix A.

**Score Function.** The score function for $k$-fingered grasps, aggregating distances between contact points and clipped to the range of $[0,1]$, is defined as $S(\boldsymbol{o}',\boldsymbol{o}) = \max\left(1 - \frac{1}{k}\sum_{j=1}^{k}||p_j^{(\boldsymbol{o}')} - p_j^{(\boldsymbol{o})}||, 0\right)$, where $p_j^{(\boldsymbol{o})}$ is the $j$-th contact point of observation $\boldsymbol{o}$.

**Inferring Object Shapes by Grasping: Results**

We adapt the distribution of airplane models by obtaining a single observation $\boldsymbol{o}$ representing contact points of $k = 5$ robot fingers with an unknown object. To evaluate the tuning process, we sample 49 objects $\boldsymbol{x}_i$ from the prior and another 49 from the posterior, and calculate the average score for the matching observations $\boldsymbol{o}_i$ obtained from the simulator using $S(\boldsymbol{o}_i,\boldsymbol{o})$. In addition, we compute the pairwise Chamfer distances between every two objects in each sampled set and take their mean as a measure of sample diversity. Scores and sample diversity are presented in Fig. 1.

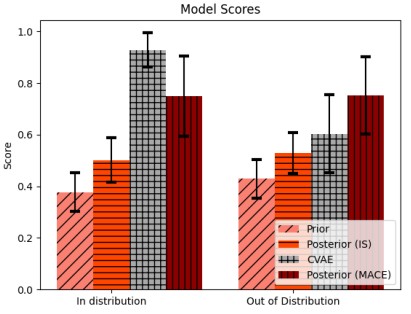

(a) Average scores

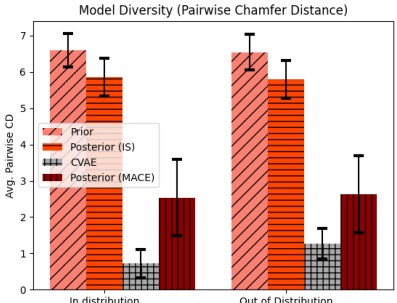

(b) Sample diversity

Figure 1: Mean results for the multi-fingered grasping experiments; error bars represent standard deviation over 100 seeds.

As a baseline, we conduct the same experiment using the importance sampling loss described in Eq. 2. We replace the log-likelihood term with the ELBO as in MACE-VAE (see Sec. 3.2.1), with the rest of the algorithm components as in Alg. 1. We find that this objective performs poorly

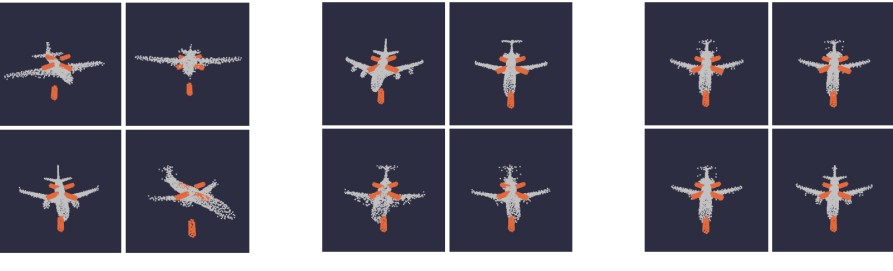

(a) Samples from the prior     (b) Samples from the posterior     (c) Samples from the CVAE

Figure 2: Tuning for the multi-finger grasping domain. Model samples are shown in white and finger positions and contact points of the given observation $o$ are represented by orange cylinders.

compared to MACE (see Fig. 1). Moreover, optimizing it is an order of magnitude (up to $20\times$) slower than using MACE.

As an additional baseline, we train a CVAE conditioned on contact points calculated by grasping each training-set object in simulation. We tune our model and compare results to the CVAE baseline using 100 objects from the held-out test set of the ShapeNet "airplane" class, all with the same observation $o$, in which the robot finger directions are the four diagonal corners of the $xy$ plane, and a fifth along the $x$ axis. Quantitative results can be seen in Fig. 1. The CVAE baseline outperforms the tuned posterior in scores, but produces a distribution with lower diversity.

In addition to the quantitative results, we present samples from the prior, posterior and CVAE models in Fig. 2 . These showcase the diversity of the model tuned by MACE compared to the CVAE distribution. Additional samples and tuning hyperparameters can be found in Appendix A.

**Out-of-distribution experiment.** Albeit its advantage of fast amortized inference, the CVAE is limited to observations seen in its training set. We demonstrate this disadvantage with an out-of-distribution experiment. We run the entire set of experiments a second time using a different observation, with the fifth finger pointing along the opposite direction of the $x$ axis. This is out of the joint distribution $p(x, o)$ which the CVAE baseline was trained on. Consequently, its results greatly deteriorate. Conversely, the model tuned by MACE outperforms it both in diversity and in scores. Results can be seen in Fig. 1. Visuals of samples from the CVAE and the MACE-tuned posterior model with the new OOD observation can be viewed in Fig. 4 in Appendix A.

## 4.2 Inverse Kinematics with Obstacles

Inverse kinematics (IK) is the calculation of the configuration of robot joints given a desired pose in Cartesian space. IK calculation is an especially challenging optimization problem when obstacles are involved and has no closed-form solution in the general case. While previous work has attempted to learn IK using generative models [28, 29, 30], we focus on tuning a pre-trained IK model to consider novel obstacles. We view the obstacle-constrained IK problem as an inference problem, where the prior $p(x; \theta_0)$ is a generative model trained to represent a distribution of joint configurations conditioned on the end-effector position[4] when no obstacles are present. Note that this is a complex and multi-modal distribution which accounts both for self-collisions and for conditioning on the desired pose. The observation is an obstacle configuration, and the posterior captures a distribution over non-colliding joint configurations.

**Dataset.** We collect 1M random valid configurations of a Franka Emika Panda 7-DoF robotic manipulator using PyBullet physics simulation [31], and record their matching end-effector positions. We collect these configurations with no obstacles present; therefore, valid configurations are ones which conform to the joint limits of the robot, and where the robot is not in collision with itself.

**Model.** We train an autoregressive model with joint positions generated sequentially, conditioned on the previous joints as well as the end-effector position: $p(q_1|\boldsymbol{p}_{ee})$, $p(q_2|q_1, \boldsymbol{p}_{ee})$ etc. Probability

---

[4]The pose can also include the end-effector orientation; in this work we focus on position-only IK.

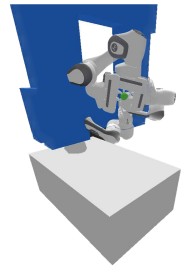 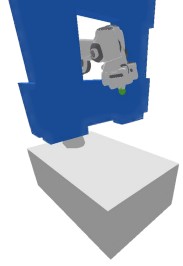 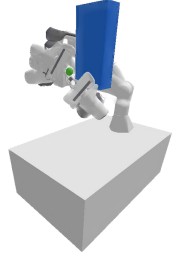 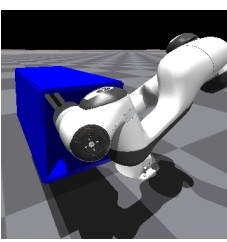

(a) Window task: prior    (b) Window task: tuned    (c) Wall task: tuned    (d) Box task

Figure 3: Tuning for the robot inverse kinematics domain. The obstacle is shown in blue, while the goal end-effector position is shown by a green ball. While Fig. 3a shows the prior distribution overlaid with the window obstacle, the same prior was used for all tasks Fig. 3b displays samples from the distribution tuned with MACE for the window obstacle, while Fig. 3c shows the same for the wall obstacle. In both cases, the posterior rarely admits configurations colliding with the obstacles, while remaining diverse.

distributions for each joint are represented by Gaussian mixture models. Further architecture and training details can be found in Appendix B.

**Simulator.** We use open-source physical simulation environments, optionally with obstacles in the robot workspace. The simulated robot can be set to a specific joint configuration $q$. The simulator returns whether the robot is in collision with itself or the obstacles, as well as the distance between the desired position and the actual end-effector position obtained by setting the robot to $q$.

**Score Function.** We opt for a score function that penalizes collisions harshly, and therefore set $S(o', o) = 0$ if the robot is in collision in a given configuration. Otherwise, the score is proportional to the distance between the generated end-effector position and the desired position: $S(o', o) = \exp(-||p_{ee,\text{desired}} - p_{ee,\text{actual}}||)$.

**Inverse Kinematics with Obstacles: Results**

We run two experiments in this domain, with different types of obstacles in the workspace, both using the PyBullet simulation environment [31]. In the first experiment, the obstacle is a vertical window, with the desired end-effector positions located beyond it. A qualitative sample from the prior model $p(x; \theta_0)$ (trained with no obstacles) can be viewed in Fig. 3a, where it is clear that many of the sampled configurations are in collision with the obstacle. Fig. 3b shows samples from the posterior model tuned with MACE, which almost never collide with the obstacle.

To show that the result does not depend on obstacle shape, we conduct a similar experiment with a wall obstacle, with the target end-effector position behind it. Samples from the model tuned by MACE can be seen in Fig. 3c, again diverse and non-colliding. We verify this

Table 1: Inverse Kinematics Results

| Model | Score | Success Rate |
|---|---|---|
| Prior | $0.129 \pm 0.015$ | $0.132 \pm 0.015$ |
| Posterior (MACE) | $0.937 \pm 0.079$ | $0.941 \pm 0.079$ |
| Posterior (IS) | $0.317 \pm 0.121$ | $0.339 \pm 0.122$ |

result quantitatively by sampling 10 goal end-effector positions behind the wall, and tuning the model with the respective score functions. As a baseline, we also tune the model with the importance sampling (IS) objective of Eq. 2. We report the mean scores over 1000 samples from the prior, the posterior tuned with MACE and the posterior tuned with the IS baseline in Table 1. We additionally report the success rate, calculated as the percentage of sampled configurations which are not in a collision state. The results clearly show improvement when tuning with MACE. Tuning hyperparameters and additional samples can be found in Appendix B.

**Comparison to MoveIt Inverse Kinematics.** The MoveIt [15] motion planning framework included with ROS has a standard IK service, used to infer goal positions for motion planning algorithms. While it is a powerful tool, we demonstrate that MACE can improve on its solutions where

Table 2: Inverse Kinematics Comparison to MoveIt for the Box Task

| MACE | | MoveIt | | MACE + MoveIt | |
|---|---|---|---|---|---|
| Time (s) | Acc. (cm) | Time (s) | Acc. (cm) | Time (s) | Acc. (cm) |
| $0.106 \pm 0.008$ | $1.92 \pm 3.01$ | $1.553 \pm 1.103$ | $< 10^{-5}$ | $0.641 \pm 0.925$ | $< 10^{-5}$ |

it struggles to find them quickly. We construct a scenario of a box in front of the robot, with the desired end-effector position inside it (see Fig. 3d). Using our prior model only (no tuning steps, for maximal speed), we sample 20 batches and test them for collisions in the IsaacGym [13] GPU-based simulator. Using our score function, we take the configurations with the maximum score as our IK solution. In Table 2, we report calculation time as well as solution accuracy for our method compared to MoveIt. In addition, since MoveIt depends on the initial robot position for the IK calculation, we use the position sampled from our model as an initial position for MoveIt, thus reaping the benefits of both methods. Time and accuracy for this setting are reported in the third column of Table 2. Experiment details (including a MACE adaptation experiment for the box domain) and additional visual results are available in Appendix B.

To alleviate **sim-to-real** concerns, we test the solutions found by MACE on a real Franka Panda robot with a box matching the IsaacGym simulator, in two different box configurations (and matching poses). Visual results of this experiment can be found at `https://www.orrkrup.com/mace`.

## 5 Limitations

**Forward simulator.** A simulator that emits observations similar to the environment may not always be available, causing a sim-to-real gap which could hurt performance. Approaches such as domain randomization [32, 18] may mitigate this problem; in addition, we show that in some cases this gap does not affect performance (see real robot experiment at `https://www.orrkrup.com/mace`).

**Inference speed.** In our experiments, MACE inference takes between $7 - 65$ seconds (depending on the domain) which is still not fast enough for real-time inference applications. While the sequential nature of MACE optimization is an unavoidable computational limitation, code optimizations as well as faster hardware[5] could dramatically speed up computation.

**Quality of the prior.** The quality of the tuned posterior depends greatly on the quality of the pre-trained deep generative model: if high-scoring samples have low probability under the prior, MACE may not find them. In our experiments, we found that deep generative models provide priors accurate enough for the domains we investigated.

## 6 Conclusion and Outlook

We presented MACE, a method for adapting deep generative models using the cross-entropy method, and demonstrated its usage for multiple robotic tasks. MACE allows the model to quickly adapt to previously unseen conditions while producing diverse posterior distributions. Our promising results for inverse kinematics show that deep generative models, when tuned appropriately using MACE, may help speed up robotic problems that are typically solved using non-learning-based approaches.

In future work, we intend to explore ways to expedite the optimization process and improve the usability of MACE in robotic tasks. Additionally, in this work we only considered the inference problem. However, in a realistic scenario the agent may also have control over *which observations to acquire*. In this case, it would be interesting to extend MACE to an active sampling method. Another related direction is to use MACE as an inference method for meta-RL, replacing the currently dominating CVAE-based approaches [12, 34].

---

[5]our experiments used unoptimized PyTorch [33] code and a single Nvidia GTX 1080 Ti GPU. As a simple proof of concept, we also ran an instance of the multi-finger grasping experiment (Sec. 4.1) on a single Nvidia A100 GPU, which cut tuning time from 25 seconds to 11 seconds.

**Acknowledgments**

The authors would like to thank Tal Daniel for sharing his considerable array of VAE training tips and tricks. In addition, we acknowledge the anonymous reviewers of the paper for their thoughtful comments and suggestions for improvement. This work was funded by the European Union (ERC, Bayes-RL, Project Number 101041250). Views and opinions expressed are however those of the author(s) only and do not necessarily reflect those of the European Union or the European Research Council Executive Agency. Neither the European Union nor the granting authority can be held responsible for them.

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

# Appendix

## A   Details for the Object Shape Inference Domain

**Prior VAE Training Details**

The prior used in this domain is a VAE trained on 2048-point PCs of ShapeNet "airplane" objects. As it is based on the architecture used in Daniel and Tamar [27], we refer the reader to their work for architectural details[6]. We train the VAE for 2000 iterations, augmenting the dataset with random rotations around the vertical ($z$) axis in the range of $\left[-\frac{\pi}{4}, \frac{\pi}{4}\right]$. Both the encoder and decoder are trained with the Adam optimizer [35], with a learning rate of $0.0005$ and a batch size of $64$. The prior standard deviation is set to $\sigma_z = 0.2$, and the weighting parameters for the loss are set to $\beta_{rec} = 50, \beta_{KL} = 1$. The latent space dimension is 128.

**Grasping Simulator Implementation Details**

To simplify implementation, we use a hand-crafted geometric simulator to calculate contact points between a theoretical robot hand and object point clouds (PCs). We assume each finger is moved along a vector pointing at the origin (which is located inside the object PC), and mark the point in the PC furthest from the origin along this vector direction as the contact point. To simulate the width of the finger, we consider points within a certain radius around the vector for contact calculation. When grasping with $k$ fingers, the observation $o \in \mathbb{R}^{k \times 3}$ is the subset of contact points from the PC $x$.

**Tuning Experiment Details**

The prior $p(x; \theta_0)$ is tuned for 2500 gradient steps (considering the value of $M$ below, this means $T \approx 78$ in Alg. 1), which takes approximately 25 seconds on a single Nvidia GTX 1080 Ti GPU. We resample a new batch of $N = 256$ samples from the updated model every $M = 32$ gradient steps, each taken on half of the batch due to memory constraints. We use the Adam optimizer with learning rate $0.0002$. We calculate the optimization objective with a quantile value of $q = \frac{1}{16}$.

**CVAE Baseline Hyperparameters**

The CVAE baseline uses an architecture similar to the VAE prior model described above, with an additional encoder to encode the condition contact points to a 128 dimensional latent $\mu_{prior}, \sigma_{prior}$. In addition to its usage in the KL divergence loss, the prior mean $\mu_{prior}$ is also used as input to the decoder (concatenated to the latent noise sampled). The CVAE baseline is trained with the same hyperparameters as the VAE described above, with two differences: $\beta_{KL} = 1$ and a learning rate of $0.0002$.

**Out-of-Distribution Experiment Visuals**

Fig. 4 shows visual samples from the MACE-tuned prior and from the CVAE in the OOD experiment (with the observation constituting a condition out of the distribution the CVAE was trained on).

**Additional Model Samples**

In this section we display additional samples for all of the distributions discussed in Sec. 4.1. All visuals follow the same color scheme as in the main text, with samples shown in white and the grasp positions represented by orange cylinders.

---

[6]See their public code at `https://github.com/taldatech/soft-intro-vae-pytorch`; we plan to release our code publicly at a later time.

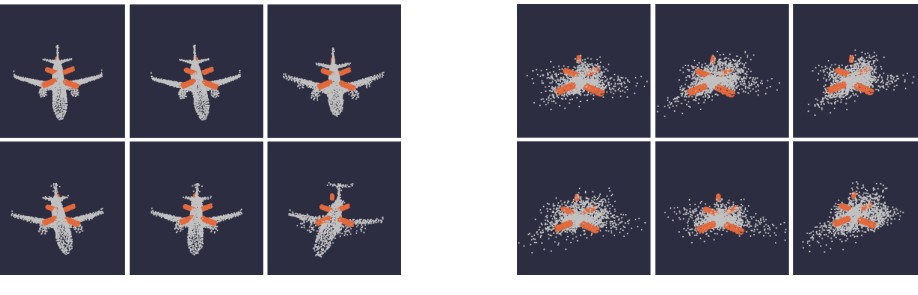

| (a) Samples from the posterior | (b) Samples from the CVAE |

Figure 4: Samples from the posterior distribution tuned by MACE (left) and the CVAE baseline (right) when using an observation that is an OOD condition for the CVAE – note the gripper finger at the tail of the airplane. Results for MACE are similar to the in-distribution task, while the CVAE is unable to generate meaningful samples.

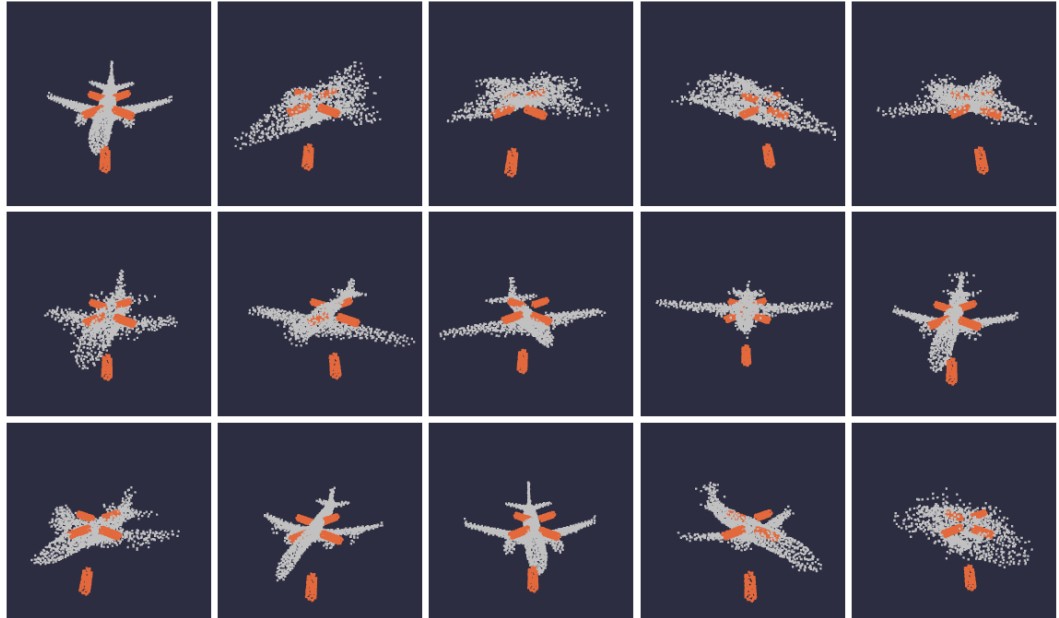

Figure 5: Samples from the prior

Fig. 5 shows samples from the pre-trained VAE prior. Figs. 6,7 display additional samples for the first (in-distribution) experiment, for the posterior tuned by MACE and the CVAE baseline respectively.

Figs. 8,9 display additional samples for the second (OOD) experiment, for the posterior tuned by MACE and the CVAE respectively.

## B Details for the Inverse Kinematics Domain

**Architecture of the Prior Model**

As mentioned in Sec. 4.2, we train an autoregressive model to produce joint configurations conditioned on end-effector positions. We use 10M data points collected using the PyBullet simulator, and train the model end-to-end with the Adam optimizer in a supervised manner, using a maximum-likelihood objective over joint configurations.. Joint probabilities are represented by Gaussian mix-

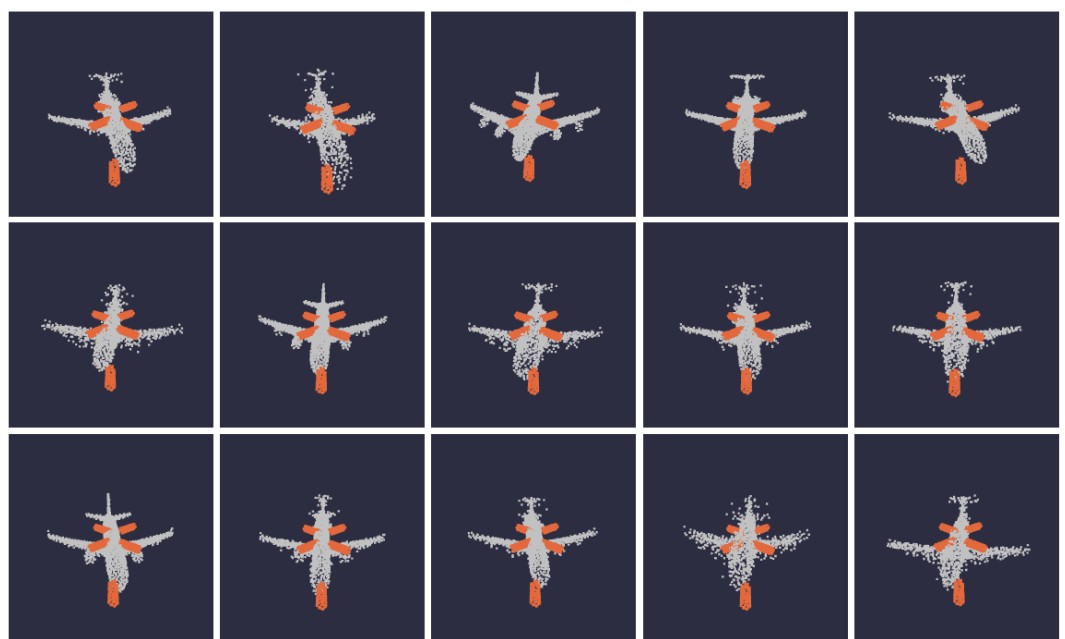

Figure 6: Samples from the posterior in the in-distribution experiment

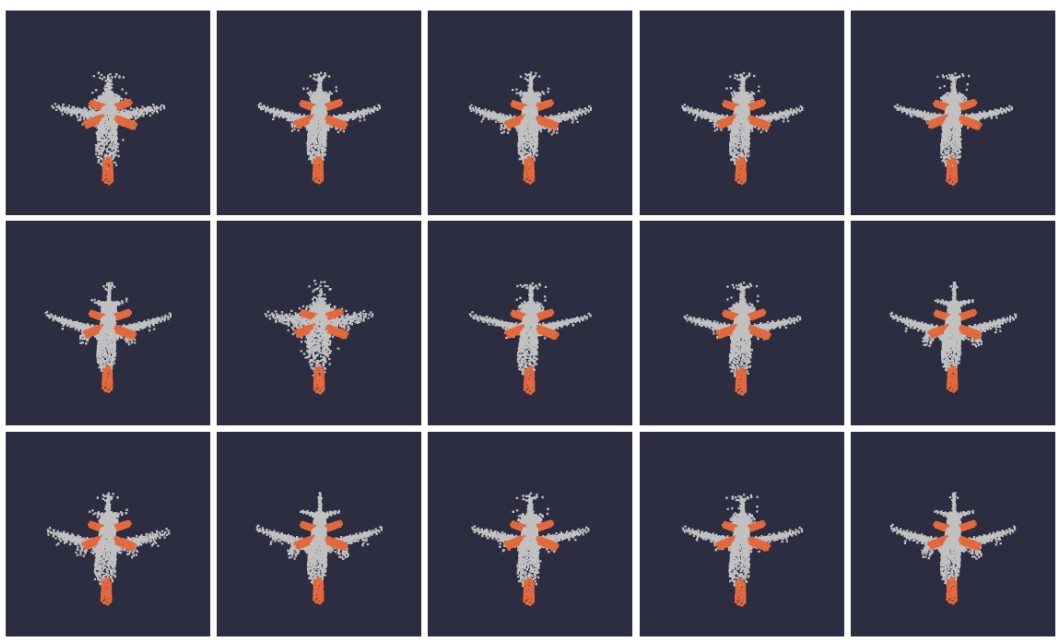

Figure 7: Samples from the CVAE in the in-distribution experiment

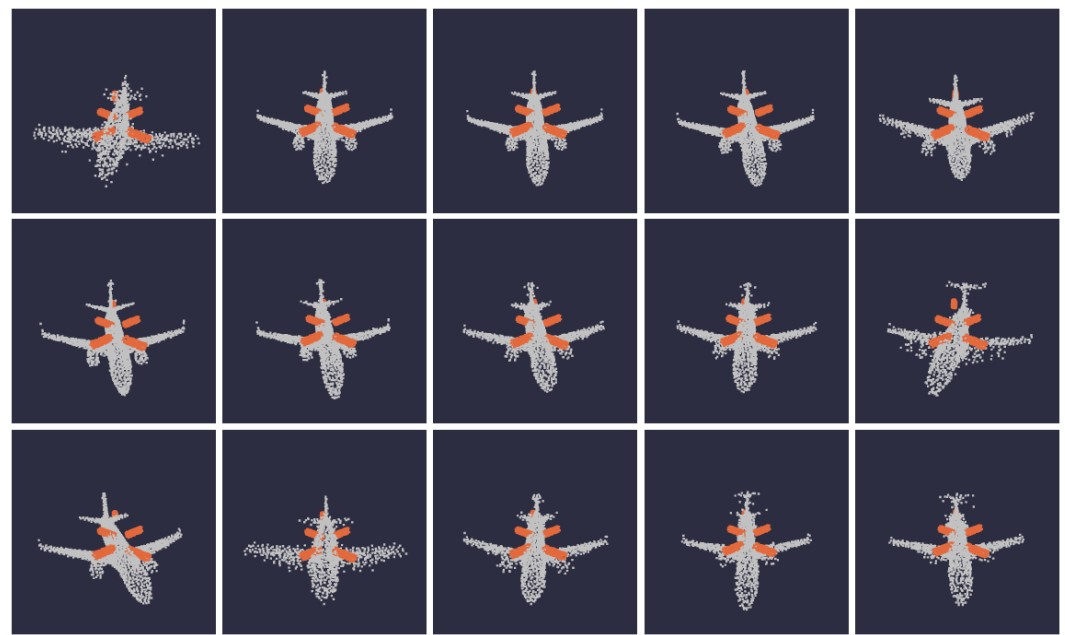

Figure 8: Samples from the posterior in the OOD experiment

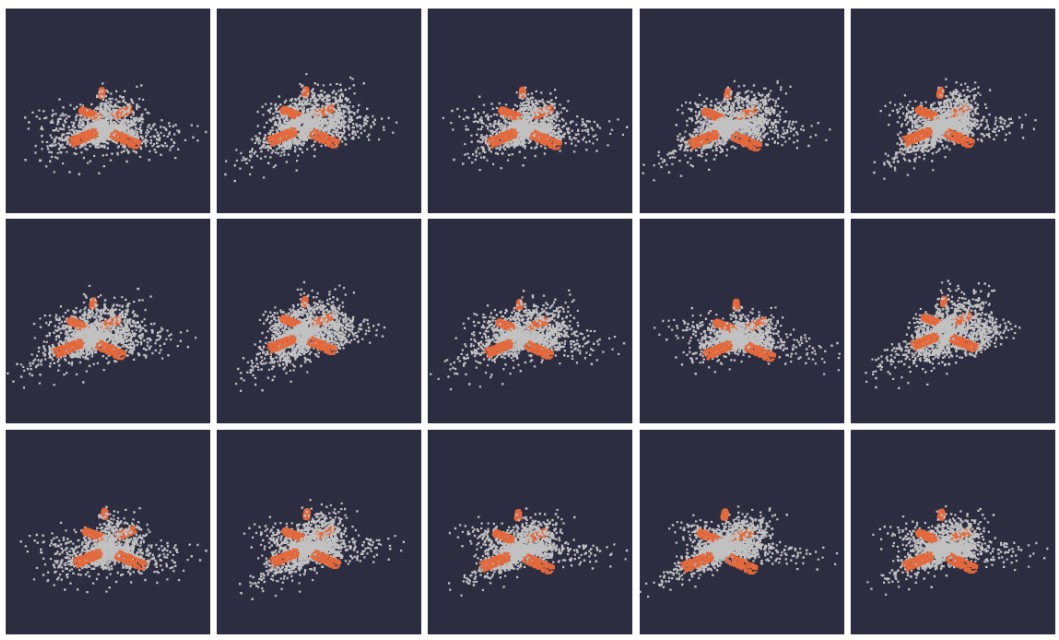

Figure 9: Samples from the CVAE in the OOD experiment

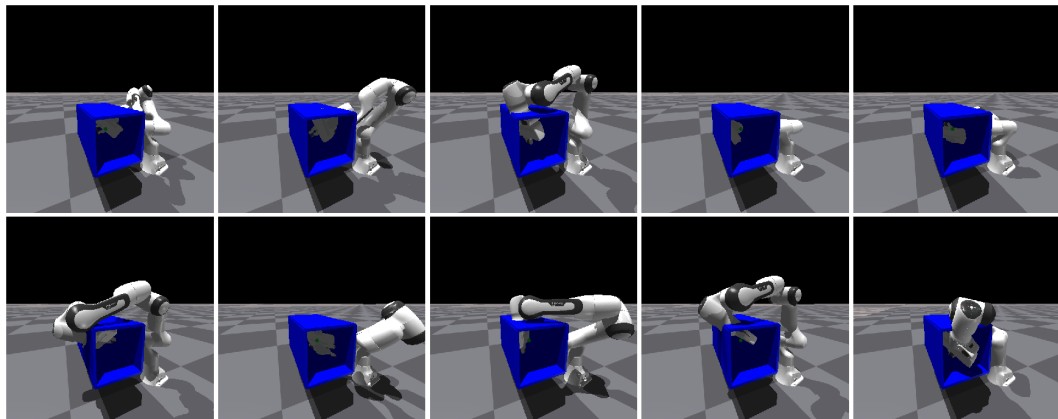

Figure 10: Samples from the prior overlayed with the box obstacle. Many of them collide with the walls of the box.

### Experiment Details

**PyBullet Experiments.** The first two experiments described in Sec. 4.2 are conducted with the PyBullet physics simulation environment, with the wall and window obstacles. We tune the pre-trained prior for 1500 fine-tuning steps (which is equivalent to $T = 375$ in Alg. 1, considering the value of $M$ below), which takes approximately 65 seconds on a single Nvidia GTX 1080 Ti GPU. We resample a new batch of $N = 64$ samples from the updated model every $M = 4$ gradient steps. We use the Adam optimizer with learning rate 0.00002. We calculate the optimization objective with a quantile value of $q = \frac{1}{16}$.

**MoveIt and IsaacGym Experiment.** For the box environment experiment and comparison to MoveIt, we use the same prior model, but instead use the GPU-based IsaacGym simulation environment to expedite scoring the samples. To calculate the results described in Table 2 of Sec. 4.2, we sample 20 batches of 4096 configurations each, and test them for collisions in IsaacGym. Obtaining the scores, we select the best configurations and report the mean and standard deviation of their distances from the goal in the accuracy column. The same configurations are used as initial positions for the "MACE + MoveIt" method in the third column, with the time constituting the total duration of sampling, testing for collisions with IsaacGym and finding solutions with MoveIt. The middle column reports times for MoveIt with a standard initial position. As MoveIt explicitly solves an optimization problem for the IK, its accuracy is very high; however, in some cases it takes much longer to find valid solutions.

**Tuning Experiment for the Box Domain.** In addition to the timing experiment, we conduct a tuning experiment with MACE on the box domain using IsaacGym. The experimental procedure is similar to the PyBullet experiments. We tune the model for 500 tuning steps (with $M = 4$, this means $T = 125$ in Alg. 1), taking approximately 10 seconds on a single Nvidia GTX 1080 Ti GPU with the faster IsaacGym simulator. We resample a batch of $N = 4096$ configurations every $M = 4$ gradient steps, and use a quantile of $q = \frac{1}{128}$. Fine-tuning is conducted using the Adam optimizer, with a learning rate of 0.0001. Samples from the prior can be found in Fig. 10, while samples from the tuned model can be seen in Fig. 11.

### Additional Model Samples for the PyBullet Experiments

In this section, we provide additional samples for the distributions described in Sec. 4.2. Fig. 12 and Fig. 14 provide samples from the prior model, trained with no obstacles present in the workspace.

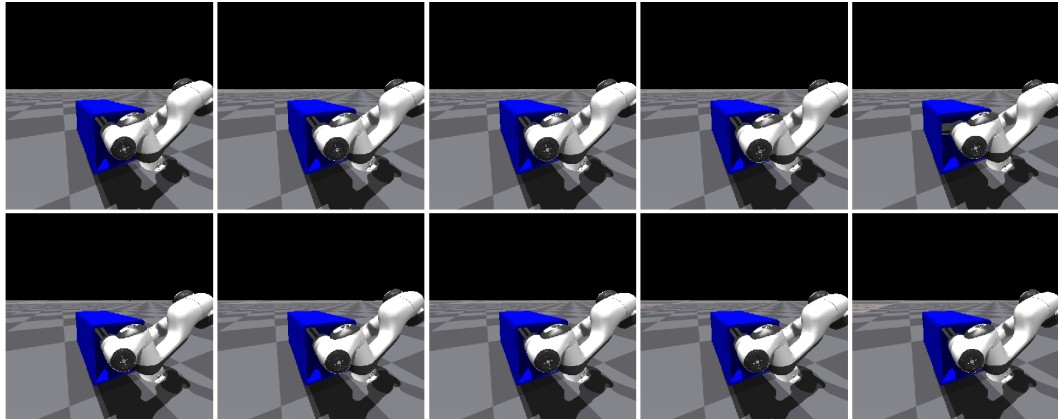

Figure 11: Samples from the posterior tuned with MACE to match the box obstacle.

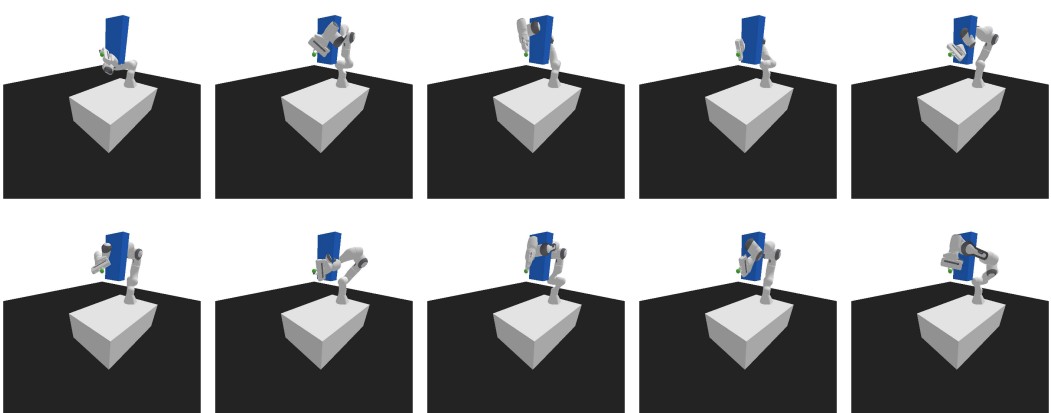

Figure 12: Samples from the prior overlayed with the wall obstacle.

This is the same distribution in both sets of samples, overlayed with different objects to show that many configurations collide with each of them.

Fig. 13 shows samples from the posterior tuned with MACE in the presence of the wall obstacle. Fig. 15 shows samples from the posterior tuned with MACE and the window obstacle.

## C  The the Point Cloud Completion Domain

PC completion is an important component of manipulation pipelines, which allows robots to reason about their environment when partial information is available from sources such as depth sensors [36, 37]. Previous work typically focuses on scenarios in which a model can be faithfully recovered given the partial PC, i.e., when the dataset is small or the partial information is indicative of the object [38, 39, 40]. Instead, we consider a case where the posterior can be extremely multi-modal, and must therefore model a highly diverse distribution.

Given a partial PC as the observation $o$, we infer a posterior distribution over possible full PCs $x$. We include this domain as a proof-of-concept, and present qualitative results on a relatively simple dataset.

**Dataset.** We use a dataset of 10K symmetrical 3D boxes generated with random edge lengths, placed on the $xy$ plane and centered around the $z$ axis. Each PC consists of 2048 points, uniformly sampled on the box faces.

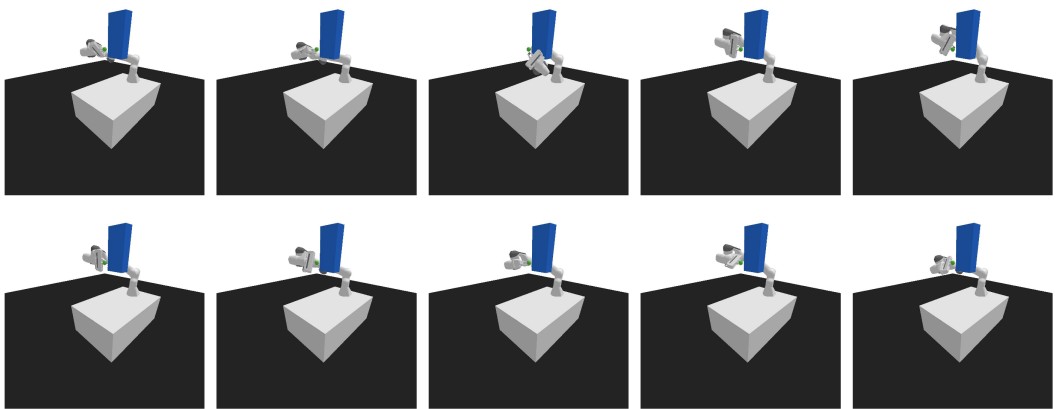

Figure 13: Samples from the posterior tuned to match observations of the wall obstacle.

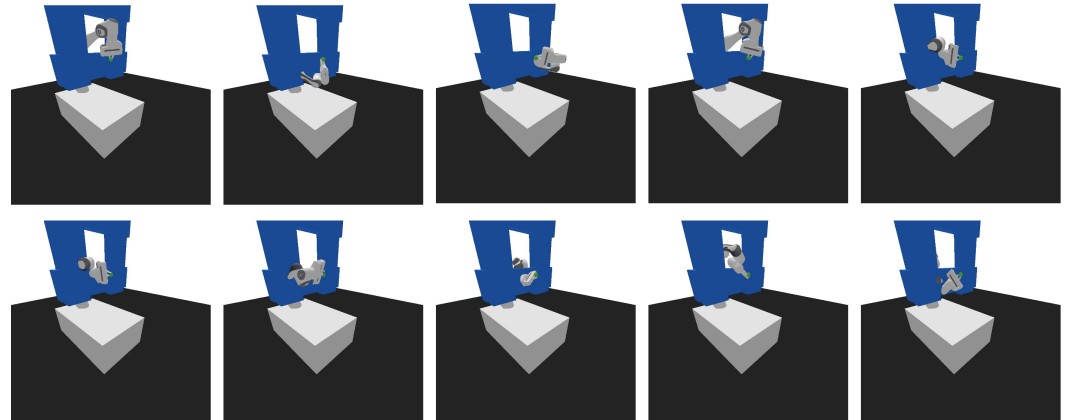

Figure 14: Samples from the prior overlayed with the window obstacle.

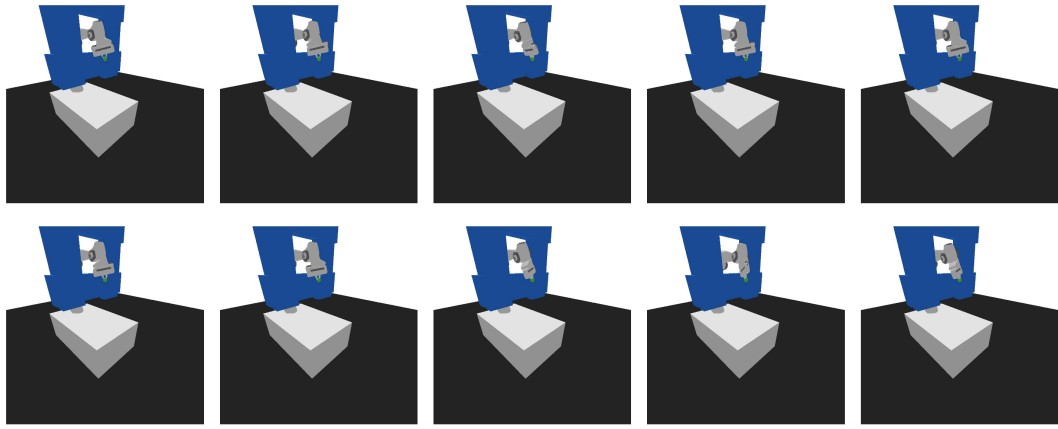

Figure 15: Samples from the posterior tuned to match observations of the window obstacle.

**Model.** We use a the same VAE architecture described in Sec. 4.1. The VAE is trained for 2000 iterations with training samples augmented by random rotation around the vertical ($z$) axis in the range of $[-\pi, \pi]$. Both the encoder and decoder are trained with the Adam optimizer [35], with a learning rate of $0.0002$ and a batch size of 64. The prior standard deviation is set to $\sigma_z = 1$, and the weighting parameters for the loss are set to $\beta_{rec} = 1, \beta_{KL} = 0.1$. The latent space dimension is 128.

**Simulator.** We require a simulator that can produce partial PCs given a full PC model. For this simple dataset, we obtain partial PCs by applying a random cut to each box, using a randomly sampled hyperplane. Note that this shape of the partial PC can fit a variety of different boxes, leading to a diverse posterior.

**Score function.** To measure similarity between PCs, $S(\boldsymbol{o}', \boldsymbol{o})$ is calculated using the Chamfer distance between PCs $\boldsymbol{o}'$ and $\boldsymbol{o}$. As suggested by Chen et al. [37], we find that calculating the distance to the top $k > 1$ nearest points produces better results than $k = 1$, and therefore use $k = 5$ when calculating the score function. Considering PCs $\boldsymbol{x}$ and $\boldsymbol{x}'$ with points labeled as $\{p_i\}_{i=1}^{N}$ and $\{p_i'\}_{i=1}^{M}$ respectively, the original Chamfer distance is given by:

$$\text{CD} = \sum_{i=1}^{N} \min_{p_i' \in \boldsymbol{x}'} ||p_i' - p_i||_2^2 + \sum_{i=1}^{M} \min_{p_i \in \boldsymbol{x}} ||p_i - p_i'||_2^2.$$

The $k$-wise Chamfer distance replaces the $\min$ operation with a selection of the top-$k$ nearest neighbors, denoted by the sets $\boldsymbol{x}^{(k)}$ and $\boldsymbol{x}'^{(k)}$:

$$\text{CD}_k = \frac{1}{k} \sum_{i=1}^{N} \sum_{p_i' \in \boldsymbol{x}'^{(k)}} ||p_i' - p_i||_2^2 + \frac{1}{k} \sum_{i=1}^{M} \sum_{p_i \in \boldsymbol{x}^{(k)}} ||p_i - p_i'||_2^2$$

To obtain scores in $[0, 1]$ with 1 being the maximum score, we set $S(\boldsymbol{o}', \boldsymbol{o}) = \exp(-\tau \text{CD}_k(\boldsymbol{o}', \boldsymbol{o}))$, where $\tau$ is a temperature parameter, set to $\tau = 0.1$ in our experiments.

**Point Cloud Completion: Results**

We use MACE-VAE (see Sec. 3.2.1) to tune the prior distribution parameters of the VAE latent space for 4000 fine-tuning steps (in Alg. 1, $T \approx 31$), which take approximately 40 seconds on a single Nvidia GTX 1080 Ti GPU. We resample a new batch of $N = 256$ samples from the updated model every $M = 128$ gradient steps, each taken on a batch of half of the samples. We use the Adam optimizer with learning rate 0.001. We calculate the optimization objective with a quantile value of $q = \frac{1}{32}$. Fig. 16a shows samples from the prior distribution $p(\boldsymbol{x}; \boldsymbol{\theta}_0)$ overlayed with the partial PC observation $\boldsymbol{o}$, while fig. 16b shows samples from the posterior model $P(\boldsymbol{x}; \boldsymbol{\theta}_T)$ tuned with MACE. We observe that MACE can produce diverse completions of the partial PC. Additional samples can be found in Figures 17,18.

**Additional Model Samples**

Fig. 17 shows samples from the pre-trained VAE prior. Fig. 18 displays additional samples for the posterior tuned by MACE.

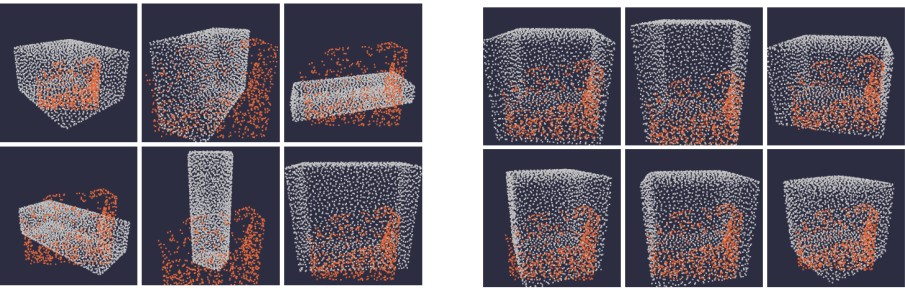

(a) Samples from the prior             (b) Samples from the posterior

Figure 16: Tuning for the PC completion domain. Samples from the prior and posterior models are shown in white. Partial PC observation $o$ is overlayed over all samples in orange. While the prior model is extremely diverse and exhibits many different box sizes and rotations, the posterior tuned with MACE converges to samples which more closely match the evidence, while still producing a plausible distribution of objects.

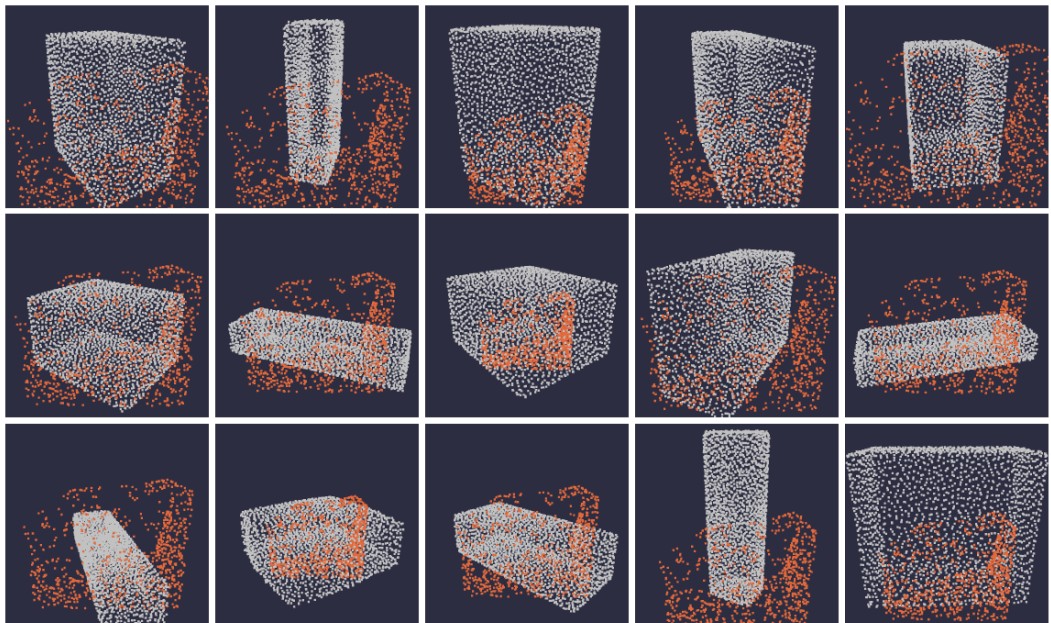

Figure 17: Samples from the prior

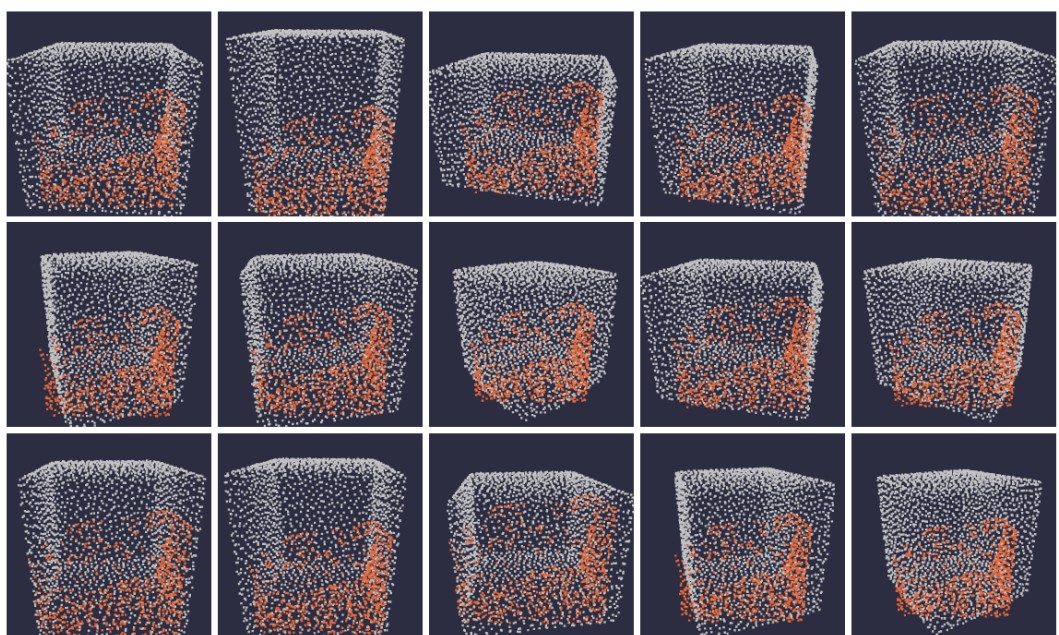

Figure 18: Samples from the posterior

