# OpenReview forum: "Fine-Tuning Generative Models as an Inference Method for Robotic Tasks"
_robot-learning.org/CoRL/2023/Conference — CoRL 2023 Poster_

### Official Review · Reviewer_LQd2 · 2023-07-17

**Confidence:** 3
**Originality:** Good
**Technical Quality:** Very Good
**Clarity Of Presentation:** Very Good
**Impact:** 4

**Recommendation:**

Weak Accept: I recommend accepting the paper, but will not argue for my recommendation if the majority of other reviewers have a different opinion.

**Review:**

Originality:
The approach of iteratively fine-tuning deep generative models using the cross-entropy method is novel. Related work on Bayes-adaptive RL is cited, but MACE differs in focusing on posterior inference rather than policy optimization. Its model-agnostic nature and reliance on simulation rather than true labels also distinguishes it from prior work.

Quality:
The method is technically sound, with a clearly motivated objective function and update rule. The experiments thoroughly evaluate both quantitative metrics and qualitative samples from MACE across multiple domains. Limitations are discussed, including inference speed and reliance on simulation. The writing clearly separates proposed contributions from analysis of results.

Clarity:
The method is technically sound with a clearly motivated objective function. Appendix provides helpful details on model architectures, training procedures and hyperparameters. Thorough experiments evaluate quantitative metrics and sample quality across domains. Limitations around inference speed and simulation fidelity are discussed.

Significance:
Efficient adaptation of internal models is an important capability for real-world robot learning. MACE provides a simple and general approach to achieve this goal for complex generative models. The method's flexibility across different models, tasks, and observation modalities demonstrates its usefulness for robotics researchers.

Relevance:
The paper is relevant to the CoRL audience. It addresses a core robot learning problem using generative models, which are seeing increasing use for robotics. The experiments cover simulation and real-world tasks like grasping.

Limitations:
The limitations around inference speed and simulation quality are adequately discussed. Testing on more complex observation spaces could further validate the generality.

**Quality Of The Limitations Section:**

Limitations are addressed clearly

**Questions For Rebuttal:**

Some potential issues that could be addressed for the revision:

* Provide more intuition and analysis on why cross-entropy optimization is effective for this problem setting
* Expand the discussion of simulation accuracy and potential impacts on real-world performance
* Test on more complex observation spaces to further validate generality
* Demonstrate on real robotic platforms beyond just simulation experiments
* Investigate ways to improve inference speed through optimizations and hardware
* Consider theoretical analysis of the properties of the iterative fine-tuning procedure
* Release code and datasets to allow for easier adoption and validation

**Robotics Focus:**

Highly relevant to robotics but no hardware experiments

**Summary Of Paper:**

The paper proposes a new method called MACE for efficiently adapting deep generative models to new observations or evidence from the environment. The key idea is to frame model adaptation as fitting the model distribution to the posterior given observations, and use an iterative cross-entropy-based fine-tuning approach (called MACE) to update the model parameters. MACE is demonstrated on several robotics domains including shape completion, tactile sensing for grasping, and inverse kinematics with obstacles. Experiments show it can produce accurate and diverse posteriors adapted to observations across these tasks.

**Summary Of Recommendation:**

The paper presents a novel method for efficient adaptation of generative models that has the potential for high impact in robotics. Experiments across diverse simulation domains demonstrate the approach can produce accurate and diverse posteriors adapted to observations. While relying on simulation and requiring optimization, the general framework offers useful tools for enabling robots to update mental models online.

---

### Official Review · Reviewer_KQ64 · 2023-07-19

**Confidence:** 4
**Originality:** Good
**Technical Quality:** Good
**Clarity Of Presentation:** Very Good
**Impact:** 2

**Recommendation:**

Weak Reject: I recommend rejecting the paper, but will not argue for my recommendation if the majority of other reviewers have a different opinion.

**Review:**

I noted the following strengths:

- the paper is generally polished very well. Especially the structure of the method section, and algorithm 1 are clear and easy to understand.

- coping with out-of-distribution scenarios is interesting, and therefore, the benefits over CVAE are made clear.

- adapting existing bayesian inference methods for robotic tasks, is a relevant topic at CoRL. The ideas of using simulation to perform bayesian inference are also trendy topic in robotics as well.

On the other hand, I noted the following potential shortcomings:

- Several assumptions made during derivations, may need better justifications either empirically or theoretically.

For example, the assumption that the simulators are deterministic and thus the expectation is replace with a single term, the assumption that the importance sampling term in equation 2 can be simply removed, the assumption that the score function is an estimate of the observation given tasks, etc, should be better justified. Could be interesting to see the influence of these assumptions in the experiments, and further, explain in more depth, why these can be justified.

- Experimental results are sometimes confusing. I also wonder if there was an option to use more standardized benchmarks for generative models in robotics. There is no hardware experiments, and thereby, the validation that this method would work in real world is missing.

For example, in figure 1, CVAE significantly outperforms MACE for in-distribution scenarios. The relevance of sample diversity might be explained more intuitively.  In table 2, MACE alone is significantly less accurate than MoveIT. Given that no standardized benchmark tasks are considered to compare the generative models, some hardware experiments, validating the method in real world setting would make the paper stronger. The last point is especially interesting, given that the paper builds on simulation based inference methods, and there might be needed interplay between simulation and real world.


**Quality Of The Limitations Section:**

Limitations are addressed clearly

**Questions For Rebuttal:**

See the section above.

**Robotics Focus:**

Relevant but unlikely to deploy to hardware in near future

**Summary Of Paper:**

This paper presents a framework called MACE (Model Adaptation with the Cross Entropy Method). The goal of MACE is to adapt the parameters of generative models. Using so-called cross entropy method, MACE fine tunes deep generative models. Forward simulations of the environments and some approximations of similarity functions as probability distributions, MACE provides a method to use available simulators in robotics, for the purpose of fine tuning deep generative models via approximate Bayesian inference. Two robotic experiments are considered in simulation/dataset. First is inferring object shapes by grasping, and another one is inverse kinematics with obstacles. Experiments demonstrate versatility and usability of the proposed framework for certain robotic tasks in simulation.

**Summary Of Recommendation:**

My recommendation is a weak reject. There are rooms for significant improvements regarding the derivations made in the proposed method. Solid justification of each steps and assumptions would be needed, as typical in Bayesian inference papers. Also, there is no "wow factors" in the experiments, and hardware experiments could verify that the use of simulation for the likelihood term, and the assumptions made here, if they would work in the real world.

---

### Official Review · Reviewer_EPGP · 2023-07-23

**Confidence:** 3
**Originality:** Very Good
**Technical Quality:** Good
**Clarity Of Presentation:** Very Good
**Impact:** 3

**Recommendation:**

Weak Accept: I recommend accepting the paper, but will not argue for my recommendation if the majority of other reviewers have a different opinion.

**Review:**

- The paper is well-structured and easy to read.
- The problem of quickly fine-tuning deep generative models in robotics is intriguing, and the proposed method is novel. Its effectiveness is evaluated in experiments, showing significant improvements in performance and sample diversity, especially in out-of-distribution scenarios compared to fast amortized inference.
- However, the paper lacks sufficient discussion on the trade-off between speed and performance. I raise questions about this and request further clarification from the authors.

**Quality Of The Limitations Section:**

Additional details required

**Questions For Rebuttal:**

- As the authors have acknowledged, the inference of MACE is still slower compared to real-time applications. Therefore, when considering real-world robotic applications, one could argue that adopting faster inference methods like amortized inference, even with some performance sacrifice, might be more practical. In this sense, how significant is the performance difference shown in Figure 1 for practical applications? Is the performance gap between the proposed method and amortized inference substantial, even considering the speed difference? Additionally, what are the actual inference speeds for each method?
- The proposed method involves several hyperparameters. How were they determined? Specifically, choosing values for $T$ and $M$ in Algorithm 1 seems crucial for both speed and performance. How do the results change when these values are varied?
- Minor comment: The authors mentioned, "we abuse notation by referring to $\mathbb{E}_{\boldsymbol{o}^{\prime} \sim \hat{p}(\boldsymbol{o} \mid \boldsymbol{x})} S\left(\boldsymbol{o}^{\prime}, \boldsymbol{o}\right)$ as $S\left(\boldsymbol{o}^{\prime}, \boldsymbol{o}\right)$." However, this may cause some confusion in the subsequent sample approximation (Eq. (2)) since it is not clear how $o_i$ is coming. It would be better to distinguish between these two notations.

**Robotics Focus:**

Highly relevant to robotics but no hardware experiments

**Summary Of Paper:**

This study proposes a fast and high-performance fine-tuning method for various deep generative models to adapt to robotic agents operating in the real world. The effectiveness of the method is demonstrated in tasks such as object shape inference from grasping, inverse kinematics calculation, and point cloud completion.



**Summary Of Recommendation:**

The proposal in this paper is intriguing, and I am inclined toward acceptance. However, I have concerns about the insufficient discussion regarding practical applications and the lack of thorough validation of the proposed method.

---

### Official Review · Reviewer_KLCv · 2023-07-25

**Confidence:** 4
**Originality:** Good
**Technical Quality:** Good
**Clarity Of Presentation:** Very Good
**Impact:** 4

**Recommendation:**

Weak Accept: I recommend accepting the paper, but will not argue for my recommendation if the majority of other reviewers have a different opinion.

**Review:**

The paper is mostly well written and addresses an interesting problem. However, I have a few concerns regarding details of the methodology and the experimental evaluation.

**Strengths**
* Presentation is mostly clear and well organised.
* The method is scalable to problems involving high-dimensional domains and large datasets.
* The method seems reasonably simple to implement within modern deep learning frameworks.
* Experiments are demonstrated on problems in robotics where deep generative models have shown significant potential.

**Weaknesses**
* The proposed method seems to be a straightforward application of CEM [24] to deep generative models, which would make the technical contribution somewhat incremental. The only difference is that the optimisation step involved in CEM is now replaced by stochastic gradient descent, instead of involving closed-form solutions or an application of expectation-maximisation, as in the original method.
* The method is only derived for deterministic simulations, and it's unclear if the method would be applicable and perform well with more general stochastic simulations (e.g., involving commonly present observation noise).
* The only non-trivial baseline in experiments is CVAE, while other classic approaches in statistics and ML could be applicable to these problem settings, e.g., likelihood-free inference methods (see references below).

**References**
* Greenberg, D., Nonnenmacher, M. &amp; Macke, J.. (2019). Automatic Posterior Transformation for Likelihood-Free Inference. *Proceedings of the 36th International Conference on Machine Learning*, in *Proceedings of Machine Learning Research* 97:2404-2414. Available from https://proceedings.mlr.press/v97/greenberg19a.html.
* Ong, V. M. H., Nott, D. J., Tran, M. N., Sisson, S. A., & Drovandi, C. C. (2018). Likelihood-free inference in high dimensions with synthetic likelihood. *Computational Statistics & Data Analysis*, 128, 271-291.


**Quality Of The Limitations Section:**

Limitations are addressed clearly

**Questions For Rebuttal:**

### Sec. 2
1. Regarding the claim in lines 73-75 on out-of-distribution data, it's not clear to me how the fact that MACE uses a similarity function would make a difference in these settings.

### Sec. 3.2
2. In line 115, it is assumed that the expected value of the score function w.r.t. the simulations is an estimate of the likelihood $p(\mathbf{o}|\boldsymbol{x})$. It is not clear what kind of "estimate" it refers to. Would it need to be unbiased, for example?
3. In line 118, it is stated that simulators are deterministic, which would make the likelihood no longer correspond to a conditional probability density and become degenerate, making the original Bayesian inference formulation fall apart. I was wondering if the authors would see that as something leading to complications.
4. Although inspired by Engel et al. [17], the formulation in Eq. 2 does not involve tempering as originally proposed [17, Sec. 3.1]. Have the authors considered using a tempering scheme?
5. How can the hyper-parameter $q$ be selected in general for a new problem setting?

### Sec. 4
6. The problems this method addresses seem to also fall within a likelihood-free inference framework. I was wondering if the authors considered comparisons against this type of methods, such as the ones mentioned above.

#### Sec. 4.1
7. How many training points were used to train the prior model? How many different examples was the model tested on?
8. What do the error bars in Fig. 1 correspond to (e.g., 1 std. deviation)?
9. How are samples from the posterior obtained? Are they exact samples? Or are they obtained by some approximate inference scheme?

#### Sec. 4.2
10. Are the initial IK solutions provided by MACE to MoveIt collision-free? Are they usually close to the goal configuration, given that MACE approximates the posterior?

**Robotics Focus:**

Highly relevant to robotics but no hardware experiments

**Summary Of Paper:**

This paper proposes an approach to adapt pre-trained generative models to approximate posterior distributions over task representations given new observations in robotic tasks. The approach, named model adaptation with the cross-entropy method (MACE), is based on an application of the cross-entropy method [24] to deep generative models. It is assumed that observations can be generated via deterministic simulations given samples from the generative model. A score function is then used to measure the similarity between simulated observations and the data. Implementations based on autoregressive models and VAEs are assessed in problems involving inverse kinematics and object shape inference from tactile feedback.

**Summary Of Recommendation:**

The paper addresses an interesting problem, but I have a few concerns regarding details of the methodology and experiments.

---

### Author Response · Authors · 2023-08-10
**General comments for all reviewers**

We thank the reviewers for their time, their thoughtful comments and their constructive feedback. We are glad that most of the reviewers found our ideas novel and impactful, and we hope to improve our work even further by integrating their suggestions into the final version of the paper.
Below, we address some topics discussed by multiple reviewers, and provide information about two additional experiments we’ve been able to run:

1. While we only use deterministic simulators in our work, MACE is not limited to settings where these simulators are available. As the reviewers mention in their comments, observation noise is common in many robotics simulators. With a stochastic simulator, we could sample multiple observations $o’$ for each sample $x$, and take the average in Eq. 1 over these samples as well as over the $x$ samples themselves. This would correspond to the expected loss for a sample $x$. The rest of the derivation would be need to be updated accordingly, but the final algorithmic procedure would remain similar.
2. In our derivation, we refer to the score function as an estimate of the likelihood of observations given a sample. We do not mean that this is an “estimator” in the statistical sense; rather, the score function is an approximation for the likelihood of observations given samples $p(o|x)$. When $p(o|x)$ is closely centered around a single value (eg. noisy visual observation of an object; or a categorical observation), the approximation is a good one.
3. **Real robot demonstration**: to demonstrate the usability of MACE in real-world applications, we showcase the IK inference environment using a real Franka Panda arm. Visual results can be found [here](https://sites.google.com/view/mace-fine-tuning).
4. Runtime: Multiple reviewers have raised concern over the inference time of MACE, considering it is an iterative algorithm with inherent sequential computation. While we leave algorithmic runtime improvements to future work, we run a simple test to show the potential of immediate improvement in runtime. While all the experiments in our paper were run on a single Nvidia GeForce 1080 Ti GPU, we re-ran an instance of the multi-finger grasping experiment (Sec. 4.1) on a single Nvidia A100 GPU. Inference time improved from 25 to 11 seconds for the same set of hyperparameters (T and M), without any further optimization of the code. This demonstrates the potential of speeding up inference with MACE in line with hardware advancements.

We reply to each reviewer’s additional concerns, as well as individual questions, in the official comment to each separate review below.

---

### Decision · Program_Chairs · 2023-08-30

**Decision:**

Accept (Poster)

**Comment:**

The paper presents a method to finetune a trained deep generative model with newly-obtained observation evidence. The method is demonstrated on a simulated grasping pose estimation domain and a motion planning domain both in simulation and the real world.

The reviewers have raised a number of concerns, including the assumption of having access to a deterministic sample-to-observation simulator and the unfavorable trade-off between performance and computation time. Reviewer KQ64 has also pointed out that the derivation would require further clarification.

At the same time, the reviewers have all agreed that the general problem of adapting a trained generative model to new evidence is of great interest to the community and that the proposed method is interesting and well-presented.

Therefore based on the discussion and the comments above, I recommend accepting the paper as a poster presentation. I recommend the authors to strengthen their experimental evaluation, especially towards more challenging real-world domains, and incorporate the clarification comments by the reviewers.